# Preliminary Study of Axillary Lymphatic Drainage in Cutaneous Melanoma Patients: A Cross-Sectional Study

**DOI:** 10.3390/medicina59081357

**Published:** 2023-07-25

**Authors:** Roberto Cirocchi, Michela Cicoletti, Fabrizio Arcangeli, Giovanni D. Tebala, Paolo Bruzzone, Stefano Avenia, Giulia Poli, Stefano Trastulli, Matteo Matteucci

**Affiliations:** 1Department of Medicine and Surgery, University of Perugia, 05100 Terni, Italy; roberto.cirocchi@unipg.it (R.C.); stefanoavenia1@gmail.com (S.A.); 2Dermatologic Clinic, S. Maria Hospital, University of Perugia, 05100 Terni, Italy; m.cicoletti@aospterni.it (M.C.); f.arcangeli@aospterni.it (F.A.); 3Department of Surgery, S. Maria Hospital, 06129 Perugia, Italy; gtebala@gmail.com (G.D.T.); stefano.trastulli@hotmail.it (S.T.); 4Department of Surgery, Sapienza University, 00189 Rome, Italy; 5Section of Pathology, Department of Medicine and Surgery, University of Perugia, 05100 Terni, Italy; poligiulia.mail@gmail.com; 6Department of Surgery, University of Milan, 20122 Milan, Italy

**Keywords:** melanoma, sentinel lymph node, biopsy, axilla, breast cancer

## Abstract

*Background*: The axilla is a region of fundamental importance for the implications during oncological surgery, and there are many classifications of axillary lymph node subdivision: on the basis of studies on women with breast cancer, we used Clough’s and Li’s classification. However, currently we do not have a gold-standard classification regarding axillary lymphatic drainage in melanoma patients. *Purpose*: Our aim was to evaluate how these classifications could be adapted to sentinel lymph node evaluation in skin-melanoma patients and to look for a possible correlation between the most recent classifications of axillary lymph node location and Oeslner’s classification, one of the most common anatomical classifications still widespread today. *Methods*: We analyzed data from 21 patients who underwent sentinel lymph node biopsy between January 2021 and January 2022. *Results*: Our study demonstrates that, to an extent, there is a possible difference in the use of the various classifications, hinting at possible limits of each. The data we obtained underline how cutaneous melanoma presents extremely heterogenous lymphatic drainage at the level of the axillary cavity. However, the limited data in our possession do not allow us to obtain, at the moment, results that are statistically significant, although we are continuing to enroll patients and collect data. *Conclusions*: Results of this study support the evidence that the common classifications used for breast cancer do not seem to be exhaustive. Therefore, a specific axillary lymph node classification is necessary in skin melanoma patients.

## 1. Introduction

The axilla is a complex anatomical region of fundamental importance due to its frequent implications in oncological surgery (breast cancer, melanoma and lymphomas). 

The axilla is described as a space of a pyramidal shape located between the upper arm and the side of the chest. We can distinguish the following parts: the apex (delimited medially by the first rib, antero-superiorly to the clavicle and posteriorly to the subscapular muscle), the anterior wall (formed by the pectoralis major, subclavius and pectoralis minor muscles), the posterior wall (formed by the subscapularis, teres major and latissimus dorsi muscles), the lateral wall (consisting of part of the biceps muscle and by the coracobrachialis muscle), the medial wall (formed by lateral chest wall, and covered by the upper part of serratus anterior) and the base (consisting of the axillary fascia) [1].

In the axillary adipose tissue there are arterial, venous and lymphatic vessels and the nerve bundles. 

The axillary lymph nodes, the number of which varies from 15 to 36, establish a common crossroads where the lymphatics of the upper limb and those of part of the thoracic and abdominal walls converge. 

Knowledge of the normal anatomy of the lymphatic system is very important to predict the location of lymph node metastases following primary cancer in the previously mentioned sites.

The axillary lymphatic-drainage anatomy was published in 1874 by Sappey [2].

During the dissection of corpses, he studied lymphatic drainage by injecting a mercury tracer at the level of interstitial tissue and lymphatic vessels. For over a century, Sappey’s hypotheses were accepted and, therefore, mercury was used from the 17th to the 20th century until it fell into disuse due to its toxicity [3].

In 2007, Suami and Al. developed a new method of ascertaining lymphatic drainage through the use of hydrogen peroxide and microinjections. They studied the course of lymphatic vessels from the fingertips to the axilla and showed that lymphatic collectors are located in the subcutaneous adipose tissue, adjacent to larger veins, such as cephalic and basilar veins. In addition, no communication was observed between superficial and deep lymphatics [4].

In a subsequent study, published in 2018, Suami and Al. introduced the concept of lymphosoma as a connection between the lymphatic vessels of a given body area and a corresponding group of regional lymph nodes. In this study, the skin was divided into specific territories that correlated with specific lymphatic basins [3].

There are many classifications used for the subdivision of the axillary lymph nodes.

Some are anatomical classifications, and these classifications are derived from the studies of Kirmisson [5], Poirier [6], Grossmann [7], Oelsner [8], and Pauchet and Dupret [9]. 

Others are functional or anatomical–surgical classifications, and they are described in treatises of surgical practice, such as “*Trattato di Medicina Operatoria, generale e speciale*” by F. Durante [10], “*La Pratique chirurgicale illustree*” by Pauchet [11], “*Operative Chirurgie*” by Kleinshmidt [12] or “*Noveau traité de technique chirurgicale*” written by M. Garbey, G. Ginestet and J. Pons [13].

In 1955, a classification based on the relationship between axillary lymph nodes and the pectoralis minor muscle was proposed by John W Berg [14]. Berg’s classification remained one of the most widely used surgical classifications, at least until 2010, when Clough introduced a new classification [15], followed by Li et al. in 2013 [16].

The aim of our study is the validation of these classifications of axillary lymph nodes, developed and used for breast cancer, for skin-melanoma patients with sentinel lymph nodes in the axilla region. 

## 2. Materials and Methods

This cross-sectional study gathered data on 21 patients affected by skin melanomas and axillary sentinel lymph nodes.

### 2.1. Inclusion and Exclusion Criteria

#### 2.1.1. Inclusion Criteria

Adult male and female patients who underwent, between January 2021 and January 2022, the removal of skin melanoma followed by axillary Sentinel Lymph Node Biopsy (SLNB) in the Unit of General Surgery of “S. Maria” Hospital in Terni (Terni, Italy).

#### 2.1.2. Exclusion Criteria

Cases with a stage lower than pT1b according to AJCC8th [17], or in whom lymphoscintigraphy located the sentinel lymph node out of the axillary cavity. 

### 2.2. Outcome

The aims of this study were (1) to find out if and how the classifications of Clough et al. and Li et al. could be adapted to sentinel lymph node evaluation in skin melanoma patients (2) to analyze the correlation between the location of cutaneous lesions and the intraoperative identification of axillary sentinel lymph nodes, and (3) to assess the possible correlation between the most recent classifications of the location of the axillary lymph nodes and the anatomical classifications still popular today. 

### 2.3. Interventions

#### 2.3.1. Pre-Operative Lymphoscintigraphy

Pre-operative lymphoscintigraphy was performed the same day as the surgery with an intradermal injection of 55 MBq of Technetium-99 m-nanocolloid (total volume: 0.4 mL in four intradermal deposits of 0.1 mL each on both sides of the surgical scar).

Images (anterior, lateral and oblique projections using a dual detector gamma camera) were acquired at 30 s per frame for 5 min, with a total of 10 frames, and were reviewed by the operating surgeon and the nuclear medicine specialist. 

#### 2.3.2. Wide Local Excision and Intra-Operative lymphatic Mapping

Then, we proceeded under local anesthesia with a wide, usually fusiform, local excision (WLE), removing skin and subcutaneous tissue down to the muscle fascia, whose excision usually was not required. 

The principal determinant of the appropriate excision margins was the Breslow thickness of the primary tumor (Table 1) [18].

The sentinel lymph node (SLNB) is a surgical procedure that identifies the sentinel node, the first lymph node or nodes that drain the primary melanoma site. 

Elevate radioactivity (a “hot spot”) is demonstrated by a hand-held (by the surgeon) gamma probe over the regional nodal basins: after surgical incision over this hot spot, the hand-held gamma probe helps the surgeon to find radioactive node(s); the most radioactive node or any node with a radioactive count of 10% or higher is defined the sentinel node [18].

## 3. Results

This retrospective observational study involved 21 skin melanoma (stage equal or greater than pT1b) patients with axillary sentinel lymph node(s), who underwent SLNB. 

The characteristics of the enrolled patients were as follows: -61.9% of patients included were males;-38.1% of patients were females.

The age range of patients included was 40–85. If we consider cancer staging according to pTNM classification, the data we obtained were as follows:
pT2a: 19%;pT2b: 19%;pT3a: 23.9%;pT3b: 14.3%;pT4a: 14.3%;pT4b: 9.5%.

### Data and Analysis

Skin melanoma localization was correlated to the topographical localization of the sentinel lymph node in the axilla. 

The characteristics of the enrolled patients were as follows (Figure 1 and Figure 2):
-19% of patients had a melanoma in subscapular region;-38.1% of patients with cutaneous melanoma in the upper limb;-19% of patients had cutaneous melanoma in the scapular region;-Melanoma in the pectoral/clavicular region in 9.5% of cases;-Cutaneous melanoma in the lumbar region in 9.5% of patients included in the study;-4.8% of patients with melanoma in the head/neck region.

The anatomical classification proposed by Clough et al. is based on the intersection of two landmarks: vertically, the lateral thoracic vein (LTV), tributary of the axillary vein; and horizontally, the second intercostobrachial nerve (ICBN). From the intersection of these two structures, 4 different zones can be derived:
-Zone A, adjacent to the LTV, extended from the lower axilla margin to the ICBN;-Zone B, adjacent to the LTV, extended from the ICBN to the axillary vein;-Zone C, contralateral to zone A;-Zone D, contralateral to zone B.

Based on the classification proposed by Clough et al. (Figure 3), the data we obtained were the follows:
-*Subscapular region*: 19% of all patients included in this study. In all these patients, the sentinel lymph node was in zone D.-*Upper limb*: 38.1% of patients with cutaneous melanoma in this region. In this case, we identified a heterogenous drainage. In fact, in 25% of cases the sentinel lymph node was in zone A; in 25% of patients it was in zone B. A proportion of 25% of patients had the sentinel lymph node in zone C, while the sentinel lymph node was in zone D in 25% of patients.-*Scapular region*: All these melanomas drained in zone D.-*Pectoral/clavicular region*: 9.5% of patients with melanoma in this area. These patients had their sentinel lymph node in zone D.-*Lumbar region*: 9.5% of all patients included in our study. These patients had the sentinel lymph node in zone B.-*Head-Neck*: 4.8% of melanoma in this region. This melanoma drained at zone A of the axillary cavity.

Overall, of the 21 patients considered (Figure 4)

-14.3% had axillary lymph nodes in zone A;-19% of sentinel lymph nodes were in zone B;-9.5% had axillary lymph nodes in zone C;-57.2% of patients included had the sentinel axillary lymph node in zone D.

In 2013, the group of Li et al. [16] published a study with the aim of evaluating the use of the second intercostobrachial nerve (ICBN) as a possible anatomical landmark for the dissection of the axillary lymph nodes in patients with breast cancer. Through this landmark, the authors divided the axillary space into an upper and lower part.

If we consider the classification proposed by Li et al, the data that we can obtain are as follows (Figure 5):
-Subscapular region: in all cases, the sentinel lymph node was in the upper region delimited by the second ICBN.-Upper limb: 38.1% of patients with cutaneous melanoma in this region. Of these patients, in 50% of cases, the sentinel lymph node was in the upper region delimited by the second ICBN, while in 50% of the patients, the lymph node was in the lower region bounded by this landmark.-Scapular region: in all cases considered in this study, the sentinel lymph nodes were in the upper region bounded by the second ICBN.-Pectoral/clavicular region: the melanomas in these areas made up 19% of cases, and in all patients, the axillary sentinel lymph nodes were in the upper region.-Lumbar region: only 9.5% of the melanomas were in this area. These melanomas drained in the upper region delimited by the second ICBN.-Head/Neck: only 4.8% of cutaneous melanomas were in this region and the sentinel lymph node was in the lower region bounded by the ICBN.

Overall (Figure 6),

-A total of 76.2% of patients included had an axillary sentinel lymph node located in the upper region delimited by the second ICBN.-A total of 23.8% of patients had a sentinel lymph node located in the lower region bounded by the second ICBN.

Finally, the data obtained with this study were evaluated using the Oelsner classification, one of the most common anatomical classifications used for the subdiision of axillary lymph nodes. 

According to this classification, the axillary lymph nodes can be distinguished as follows:
-Anterior or pectoral lymph nodes. These lean against the anterior dentate muscle, along the course of the lateral thoracic artery. This group receives lymph fluid that comes from the muscles of the anterosuperior wall of the chest, and from the integuments, the skin of the supra-umbilical region of the abdominal wall.-Lateral or humeral lymph nodes. These are related to the posteromedial part of the axillary vein, and they drain almost all the lymph fluid coming from the upper limb.-Subscapular or posterior lymph nodes. These are in the back of the axilla, leaning against the subscapular artery.-Apical lymph nodes (subclavicular lymph nodes). These are found in the upper portion of the axilla, posteriorly to the small pectoral muscle and medially to the axillary vein. They receive the lymph fluid collected from the other axillary lymph nodes.-Central lymph nodes. These are dispersed in the adipose tissue on the base of the axilla. They receive the lymph fluid that comes from the other axillary groups and their efferent ducts are directed to the lymph node of the apical group.

According to this classification, the data that we obtained are the following (Figure 7):
-*Pectoral (anterior) lymph nodes*: 14.3% of cases. Of these cases, in 33.3% of the patients, the cutaneous melanoma was in the head/neck region, while in 66.7% of the patients, the cutaneous melanoma was in the left upper limb.-*Apical lymph nodes*: 19% of patients had definable sentinel lymph nodes. Of these patients, 50% had melanomas in the lumbar site, while 50% had melanomas in the upper limb.-*Subscapular (posterior) lymph nodes*: only 9.5% of patients. In all cases, the cutaneous melanoma was in the upper limb.-*Humeral (lateral) lymph nodes*: 47.6% of patients had such classifiable lymph nodes. Of these patients, 40% had melanomas in the subscapular region, 40% had melanomas at the level of the scapular region, and 20% of patients had cutaneous melanomas in the upper limb.-*Central lymph nodes*: only 9.5% of the patients were included. In these cases, the cutaneous melanomas were in the pectoral/clavicular region.

Therefore, this classification was compared with the classification proposed by Clough et al.

This comparison showed the following data (Figure 8):
-The pectoral (anterior) lymph nodes corresponded to the lymph nodes which, according to the classification of Clough et al., were located in zone A;-The apical lymph nodes had a counterpart in the lymph nodes of zone B;-The subscapular (posterior) lymph nodes corresponded to the lymph nodes of zone C;-The humeral (lateral) lymph nodes corresponded to zone D;-According the classification proposed by Clough et al., there was no equivalent of central lymph nodes.

The classification proposed by Oeslner was also compared with the classification proposed by Li et al. This comparison showed the following data:
-The humeral (lateral) lymph nodes, the apical lymph nodes and the central lymph nodes were in the upper region delimited by the second ICBN;-The anterior and posterior lymph nodes were in the lower region bounded by the second ICBN.

Currently, we have not been able to find a statistically significant correlation between the site of the melanoma and the area of lymphatic drainage in the axilla. 

However, these data show how, to an extent, there are possible differences in the use of the various classifications, hinting at possible limits of using of not using them; therefore, this shows potential that, in the future continuation of this study, good results should be possible to obtain. 

## 4. Discussion

Cutaneous melanoma represents, in Italy, the second most frequent cancer in the male sex and the third most frequent tumor in the female sex. The risk of developing melanoma is 1.5% in males and 1.2% in women [19].

SLNB provides relevant prognostic information; furthermore, the positivity of the sentinel lymph nodes, together with other parameters, dictates the type of treatment to which the patient should be subjected. 

Herbert Snow realized that melanoma first spread to regional lymph nodes before metastasizing to distant sites: in 1892, he recommended elective nodal dissection (ELND) both as a curative measure and to improve regional control [20].

This recommendation remained controversial for more than a century, as a consequence of the possible and unnecessary comorbidities to which the patients could be exposed. 

These controversies were based on numerous randomized trials. 

In 1998, a World health Organization study randomized 240 patients with truncal melanoma of 1.5 mm or more in thickness to ELND or nodal observation in order to evaluate the efficacy of immediate node dissection in patients without clinical evidence of regional node and distant metastasis. It was concluded that node dissection increased survival in patients with node metastases only [21].

However, there were also numerous other studies that showed the effectiveness of axillary lymph node dissection.

In 1996, Balch et al. conducted the first randomized clinical trial that demonstrated the value of surgical treatment for clinically occult regional metastases, and also that patients 60 years of age or younger with intermediate-thickness melanomas, particularly those with nonulcerative melanomas and those with tumors 1 to 2 mm thick, could benefit from ELND [22].

However, in the absence of a real benefit, ELBN was received with little enthusiasm, especially for the comorbidities associated with this technique (chronic pain and lymphoedema). 

In 1977, Robinson et al. published a paper concerning cutaneous lymphoscintigraphy to identify the lymph nodes that drained truncal melanomas [23,24].

In 1992, Morton et al. demonstrated that SLNB was indicative of the presence or absence of microscopic lymph node metastasis [25]. Many subsequent studies validated the SLN hypothesis in a variety of malignant neoplasms, and SLNB has become the gold-standard method for regional-node staging in melanoma and breast cancer.

SLNB is not a major surgical procedure, but complications, including seroma, wound infection, nerve damage and lymphedema, sometimes occur: a study conducted by Morton et al. [26] suggest that very few patients who underwent SLNB for cutaneous melanoma had objective lymphedema in the long term.

Therefore, the reason is clear why the specific knowledge of axillary lymph node anatomy is so important.

Cirocchi et al., in 2020 [27], highlighted the importance of defining new markers to preserve as many axillary lymph nodes as possible in breast surgery.

We, here, tried to evaluate if these classifications of axillary lymph nodes, used for breast cancer, could be adequate for patients with melanoma and with sentinel lymph nodes located at the axillary level.

Clough et al. [15] reported 227 patients with stage I breast cancer (T1/T2-N0) or ductal carcinoma in situ (DCIS) who underwent SLNB. That study showed that in most cases, the sentinel lymph node was in zone A.

The results that we are presenting in this paper on skin melanoma patients disagree with those obtained by Clough et al. in breast cancer patients: in our study, only in a small percentage of cases did the patients present with sentinel lymph nodes in zone A. More frequently, the sentinel lymph nodes were at the level of zone D.

In 2013, the group of Li et al. [16] published a paper proposing the second intercostobrachial nerve (ICBN) as a possible anatomical landmark for axillary lymph node dissection in patients with breast cancer. That study showed that the sentinel lymph nodes were most frequently located below the second ICBN. Our study, once again, showed discordant results: in fact, in most cases of skin melanoma included in our study, the sentinel lymph nodes were in the region located above the second ICBN.

The results we obtained were also evaluated in light of the Oeslner classification, one of the anatomical classifications mainly used.

Each of the above classifications has critical issues. In fact, as we have mentioned, the classification of Clough et al. does not consider what, according to Oeslner’s classification, the central lymph nodes are (the site of the sentinel lymph node in 9.5% of the patients included in our study). The classification proposed by Li et al. does not allow the accurate description of the lymphatic drainage station in the axilla. Oelsner’s Classification is the most complete classification, but it is a purely anatomical classification and it is difficult to apply in surgical practice.

The data we obtained underline how cutaneous melanoma presents an extremely heterogenous lymphatic drainage at the level of the axillary cavity. A previous study by Cirocchi et al. [28] had shown how the lymphatic drainage of cutaneous melanoma is extremely heterogenous: it is more various if compared to the topographic localization of the lymph nodes at the level of the axillary cavity.

It is evident that a specific classification is necessary for axillary lymph nodes in case of cutaneous melanoma.

### Study Limitations

The limited data available to us do not presently allow us to obtain statistically significant results, although the enrollment of patients and data collection is progressing, after the end of the SARS-CoV-2 pandemic emergency, which caused, in Italy, a dramatic decline of surgical procedures.

## 5. Conclusions

Given the significant heterogeneity of the lymphatic drainage of cutaneous melanoma at the level of the axillary cavity, the common classifications used for breast cancer do not seem to be adequate, and a specific classification is necessary: in fact, the exact localization of the sentinel lymph node helps surgical dissection and reduces the morbidity associated with SLNB, and, in particular, the risk of lymphedema and sensory impairment. This is a preliminary study: the goal is to increase, in the future, the data in our possession, with the intent to identify new anatomical landmarks and to establish if and how the prognoses of patients with cutaneous melanoma varies depending on the different axillary regions in which the sentinel lymph node is located.

## Figures and Tables

**Figure 1 medicina-59-01357-f001:**
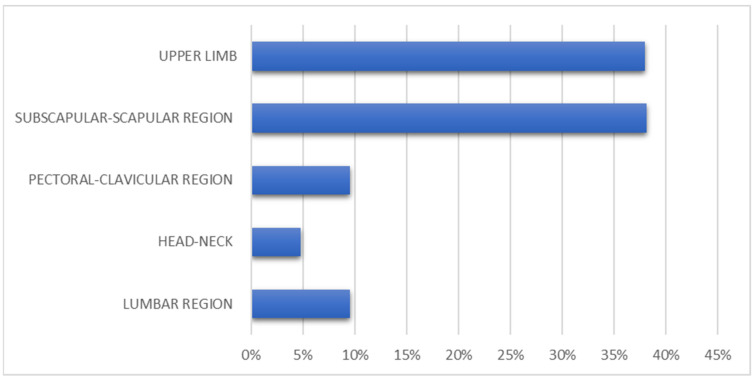
Localization and frequency of primary melanomas included in this study.

**Figure 2 medicina-59-01357-f002:**
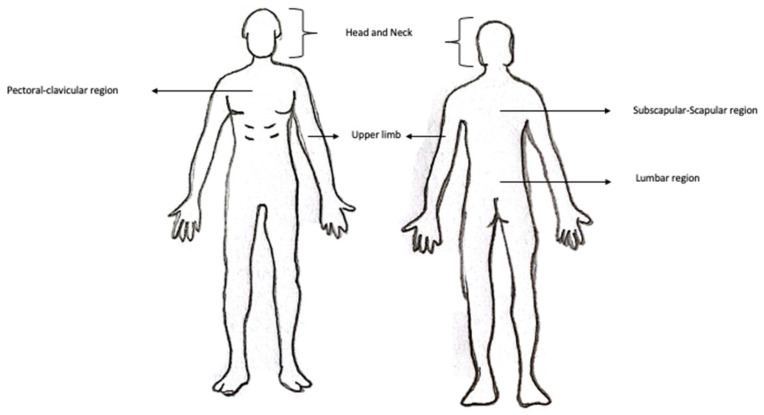
Topographic location regions of melanomas included in the study.

**Figure 3 medicina-59-01357-f003:**
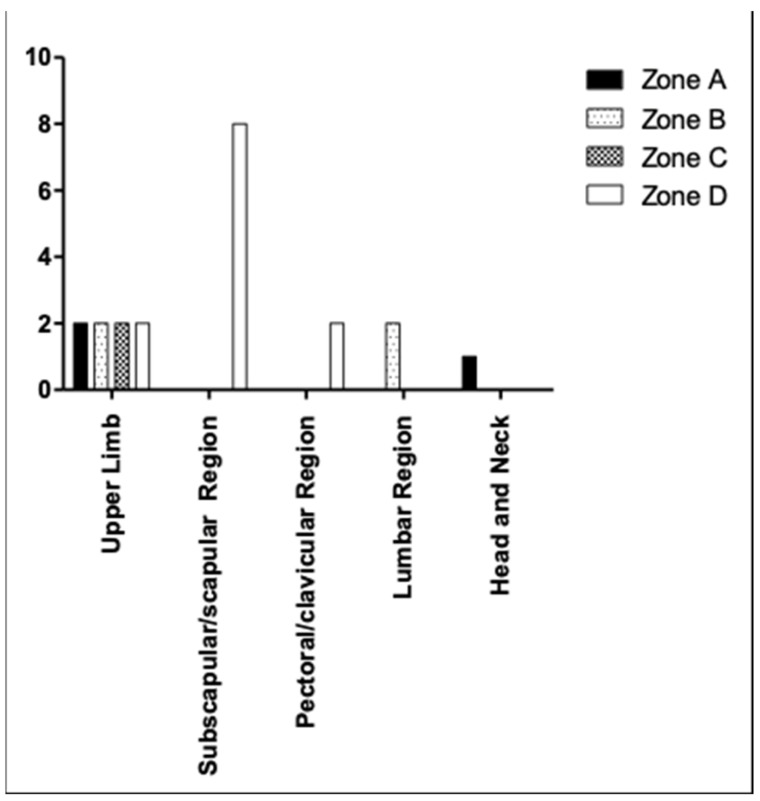
Different localization of the axillary sentinel lymph node (according to Clough’s classification) seems to be related to the different regions of onset of the primary skin melanoma.

**Figure 4 medicina-59-01357-f004:**
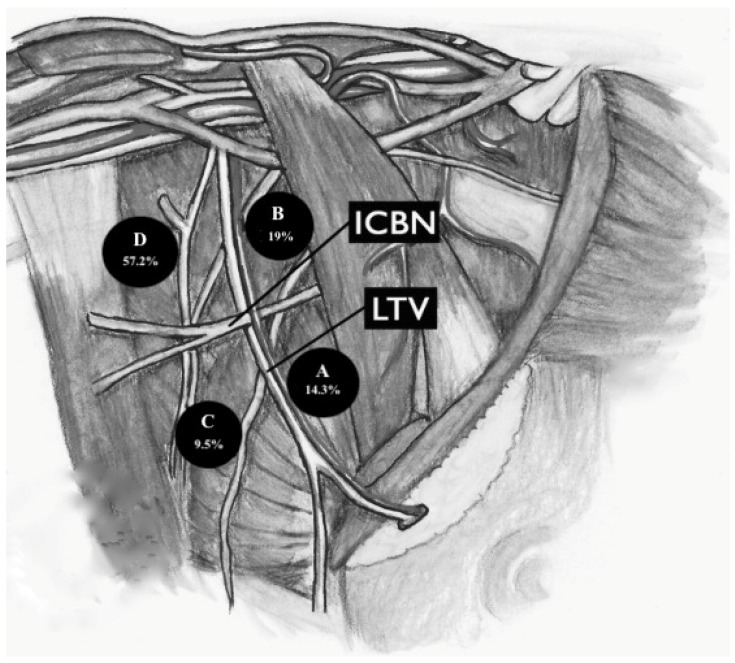
Different topographic localization of sentinel lymph node (according to the classification of Clough et al.) in relationship with the topographic localization of melanomas.

**Figure 5 medicina-59-01357-f005:**
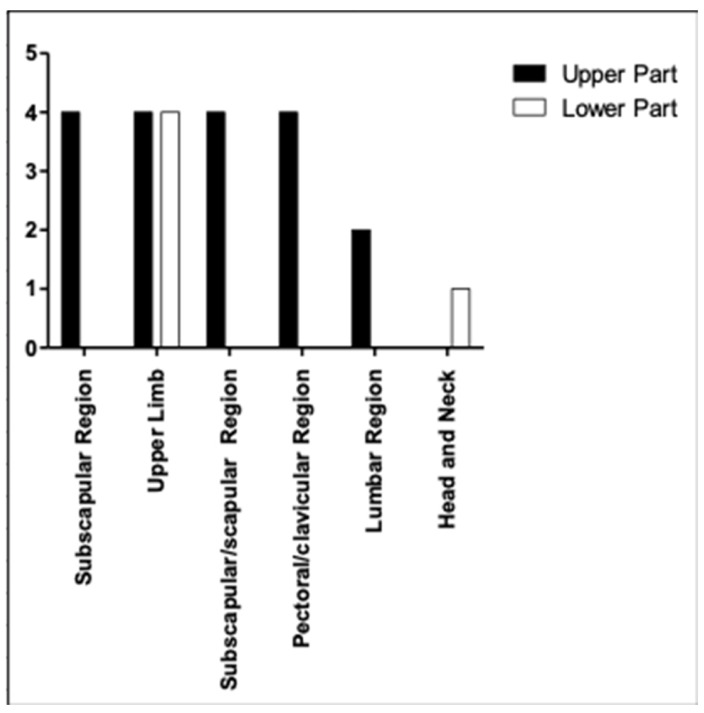
Localization of the axillary sentinel lymph node according to the Li et AL. classification. in relation to the different site of onset of primary cutaneous melanoma.

**Figure 6 medicina-59-01357-f006:**
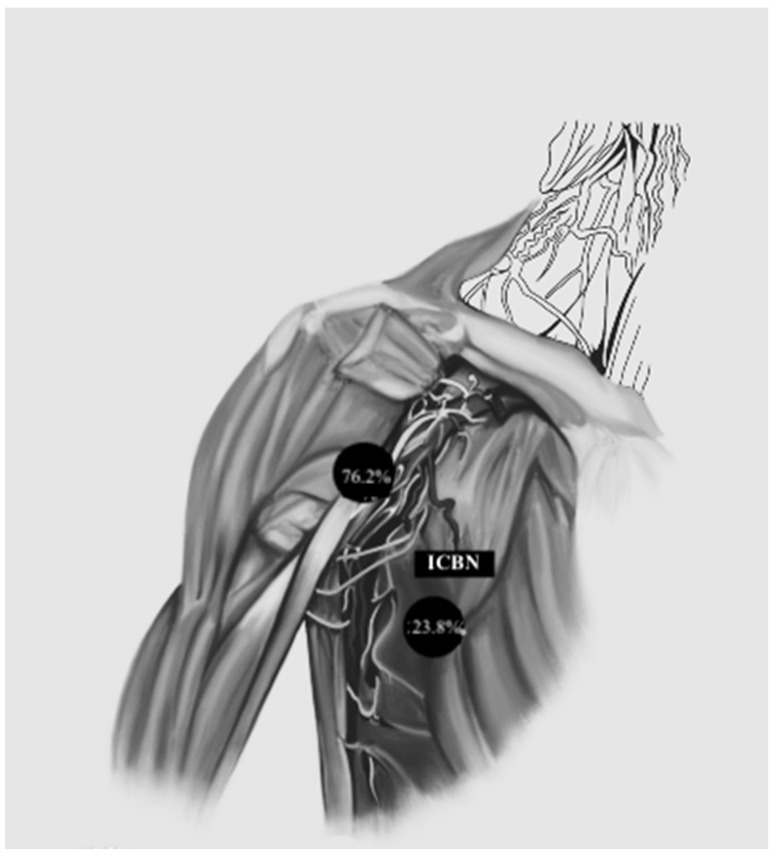
Localization of the axillary sentinel lymph node according to the classification of Li et al.

**Figure 7 medicina-59-01357-f007:**
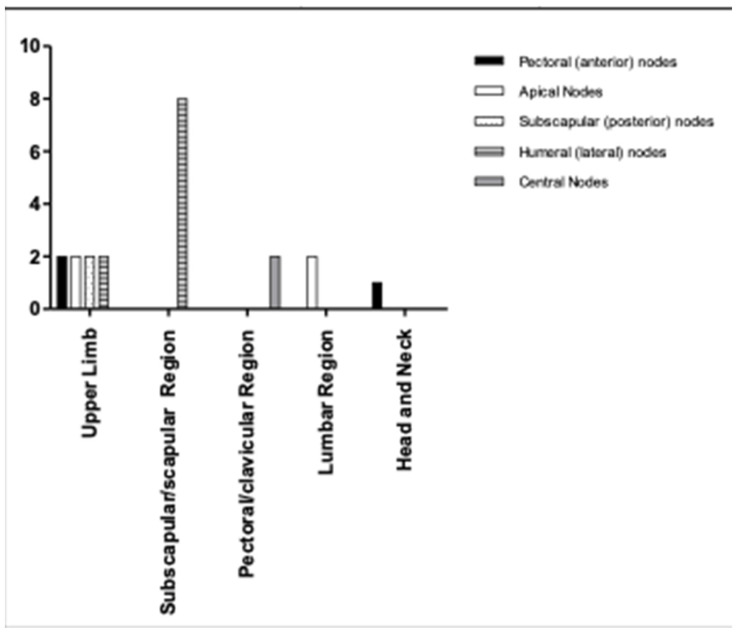
Localization of the axillary sentinel lymph node according to the Oelsner Classification in relation to the site of the primary skin melanoma.

**Figure 8 medicina-59-01357-f008:**
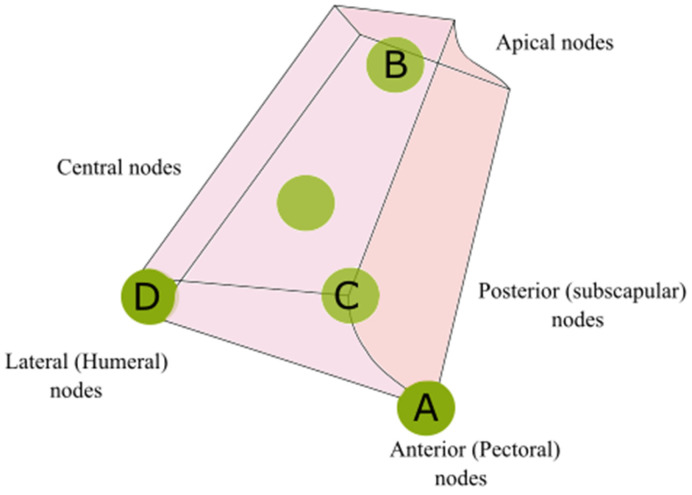
Graphical representation of the comparison between Oelsner’s classification with the classification proposed by Clough et al.

**Table 1 medicina-59-01357-t001:** Recommended margins of wide local excision.

Thickness	Margin
**Melanoma in situ**	0.5 cm
**<1 mm**	1 cm
**1–2 mm**	1–2 cm
**>2–4 mm**	2 cm
**>4 mm**	2 cm

## Data Availability

The data presented in this study are available on request.

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
