# Peer review of "Preliminary Study of Axillary Lymphatic Drainage in Cutaneous Melanoma Patients: A Cross-Sectional Study"

_medicina, 2023, doi:10.3390/medicina59081357_

Round 1
Reviewer 1 Report
I would like to thank the authors for the opportunity to review their paper about the axillary lymphatic drainage of cutaneous melanoma. Melanoma is the most aggressive skin cancer; therefore, it is necessary to make all the effort to understand the dynamic of this cancer.
Hereby please find my comments regarding the paper:
- The number of the subtitle are only 1 and 2. The material and methods have the same number with the results.
- Some abbreviations are not explained when appeared for the first time.
- Abstract: the sentences are too long; it is necessary to be more concise.
- It is currently used the term melanoma, not malignant melanoma.
- At table 1, please remove the term tumore.
- Results: all the results must be express in percentages. The results need to be reorganized; they are confusing. In figure 1 the percentage or the number of the patients are missing. Before you present the results based on each classification you need to present the classification itself, otherwise are confusing.
- The quality of the images needs to be improved.
- Discussions: no comparison can be made between 277 or 72 patients with 21 patients. Please extend your study period before or after the analyzed period.
- References: the authors should add more current references, written correctly.
I would like to thank the authors for the opportunity to review their paper about the axillary lymphatic drainage of cutaneous melanoma. Melanoma is the most aggressive skin cancer; therefore, it is necessary to make all the effort to understand the dynamic of this cancer.
Hereby please find my comments regarding the paper:
- The number of the subtitle are only 1 and 2. The material and methods have the same number with the results.
- Some abbreviations are not explained when appeared for the first time.
- Abstract: the sentences are too long; it is necessary to be more concise.
- It is currently used the term melanoma, not malignant melanoma.
- At table 1, please remove the term tumore.
- Results: all the results must be express in percentages. The results need to be reorganized; they are confusing. In figure 1 the percentage or the number of the patients are missing. Before you present the results based on each classification you need to present the classification itself, otherwise are confusing.
- The quality of the images needs to be improved.
- Discussions: no comparison can be made between 277 or 72 patients with 21 patients. Please extend your study period before or after the analyzed period.
- References: the authors should add more current references, written correctly.
Author Response
Dear Reviewer,
Thank you for your review. We have made these changes:
- Subtitle numbers have been changed
- All abbrevations are now explained when they appear for the first time
- Abstract has been changed. Now sentences are more concise.
- We removed the term malignant melanoma
- In table 1 we removed the term tumore
- All results are now expressed in percentage. Figure 1 has been changed and the results have been reorganized.
- Images have been changed.
- This is a prelimanary study. We are trying to continue to enroll patients and collect data.
- References have been modifying.
- We have added a new author who provided an editing of English Language.
King Regards
Reviewer 2 Report
Dear Editors,
This study is generally well designed. It will contribute to the literature. I have some suggestions to make it a more targeted and nice article;
1- The language of the article needs to be improved (there are grammatical errors and typos). It is recommended that the article is edited by a native English speaker.
2- The number of cases is low, this must be mentioned in the limitation part.
3- The discussion part should be deepened;
- Fee, H. J., Robinson, D. S., Sample, W. F., Graham, L. S., Holmes, E. C., & Morton, D. L. (1978). The determination of lymph shed by colloidal gold scanning in patients with malignant melanoma: a preliminary study. Surgery, 84(5), 626-632.
- Morton, R. L., Tran, A., Vessey, J. Y., Rowbotham, N., Winstanley, J., Shannon, K., ... & Saw, R. P. M. (2017). Quality of life following sentinel node biopsy for primary cutaneous melanoma: health economic implications. Annals of Surgical Oncology, 24, 2071-2079.
Best wishes ...
Dear Editors,
This study is generally well designed. It will contribute to the literature. I have some suggestions to make it a more targeted and nice article;
1- The language of the article needs to be improved (there are grammatical errors and typos). It is recommended that the article is edited by a native English speaker.
2- The number of cases is low, this must be mentioned in the limitation part.
3- The discussion part should be deepened;
- Fee, H. J., Robinson, D. S., Sample, W. F., Graham, L. S., Holmes, E. C., & Morton, D. L. (1978). The determination of lymph shed by colloidal gold scanning in patients with malignant melanoma: a preliminary study. Surgery, 84(5), 626-632.
- Morton, R. L., Tran, A., Vessey, J. Y., Rowbotham, N., Winstanley, J., Shannon, K., ... & Saw, R. P. M. (2017). Quality of life following sentinel node biopsy for primary cutaneous melanoma: health economic implications. Annals of Surgical Oncology, 24, 2071-2079.
Best wishes ...
Author Response
Dear Reviewer, thank you for you review.
We have made these changes:
- The English language has been improved from a new author
- We have added a part, titled "limit of the study"
- The discussion part has been deepened and we added the two articles suggested.
King regards
Round 2
Reviewer 1 Report
Dear Authors,
In methods from the abstract you have an editing error (were analyzed appear to the beginning and to the end of the phrase). Also you need to write the bibliography according to journal protocol.
The quality of English Language are improved.